**DOI: 10.1038/ncomms13938**　　**OPEN**

# Suppressed decomposition of organometal halide perovskites by impermeable electron-extraction layers in inverted solar cells

K.O. Brinkmann[1], J. Zhao[1,2], N. Pourdavoud[1], T. Becker[1], T. Hu[1,3], S. Olthof[4], K. Meerholz[4], L. Hoffmann[1], T. Gahlmann[1], R. Heiderhoff[1], M.F. Oszajca[5], N.A. Luechinger[5], D. Rogalla[6], Y. Chen[3], B. Cheng[2] & T. Riedl[1]

The area of thin-film photovoltaics has been overwhelmed by organometal halide perovskites. Unfortunately, serious stability concerns arise with perovskite solar cells. For example, methyl-ammonium lead iodide is known to decompose in the presence of water and, more severely, even under inert conditions at elevated temperatures. Here, we demonstrate inverted perovskite solar cells, in which the decomposition of the perovskite is significantly mitigated even at elevated temperatures. Specifically, we introduce a bilayered electron-extraction interlayer consisting of aluminium-doped zinc oxide and tin oxide. We evidence tin oxide grown by atomic layer deposition does form an outstandingly dense gas permeation barrier that effectively hinders the ingress of moisture towards the perovskite and—more importantly—it prevents the egress of decomposition products of the perovskite. Thereby, the overall decomposition of the perovskite is significantly suppressed, leading to an outstanding device stability.

[1] Institute of Electronic Devices, University of Wuppertal, Rainer-Gruenter-Str 21, 42119 Wuppertal, Germany. [2] College of Materials Science and Engineering, Nanchang University, 999 Xuefu Avenue, Nanchang 330031, China. [3] College of Chemistry/Institute of Polymers, Nanchang University, 999 Xuefu Avenue, Nanchang 330031, China. [4] Department of Chemistry, University of Cologne, Luxemburger Straße 116, 50939 Cologne, Germany. [5] Nanograde AG, Laubisrütistrasse 50, 8712 Stäfa, Switzerland. [6] Ruhr-Universität Bochum, RUBION, Universitätsstr. 150, 44801 Bochum, Germany. Correspondence and requests for materials should be addressed to T.R. (email: t.riedl@uni-wuppertal.de).

Solar cells based on organolead halide perovskites have seen tremendous progress over the past 5 years[1]. Although their power conversion efficiency (PCE) has skyrocketed to reach a level of >20% (ref. 2), concerns about their stability are still intimately linked to this technology[3–5].

Generally, there is consensus that perovskites like $CH_3NH_3PbI_3$ decompose to their constituents, that is, HI, $CH_3NH_2$ and $PbI_2$, in the presence of water[6]. Moreover, the intrinsic decomposition of $CH_3NH_3PbI_3$ to $CH_3NH_3I$ and $PbI_2$, which is thermally activated and which occurs even under inert conditions, states a more fundamental problem[7,8]. At the same time, there are secondary effects of perovskite decomposition such as the degradation of functional building blocks in the solar cell because of the corrosive effect of the perovskite decomposition products[9]. Specifically, the corrosion of metal electrodes like Ag or Al has been identified to be a critical issue[10]. Overall, without proper concepts to overcome these reliability issues, the prospects of wide-spread application and commercialization of organolead halide perovskite technology may be significantly compromised.

Among the various device architectures for perovskite solar cells, the inverted planar geometry has been shown to yield devices essentially free of hysteresis[11,12]. Here, the photogenerated holes are extracted via the bottom electrode using a hole-extraction interlayer (for example, poly(3,4-ethylene-dioxythiophene) polystyrene sulfonate (PEDOT:PSS)), while the electron extraction proceeds via the top electrode. The energetic alignment of $CH_3NH_3PbI_3$ and [6,6′]-phenyl-C61-butyric acid methyl ester (PCBM) has been shown to be especially favourable for the extraction of electrons from the conduction band of the perovskite[13]. To overcome the stability issues discussed above, several groups have introduced inverted planar perovskite cells based on metal oxides as electron- and hole-extraction interfacial layers adjacent to the cathode and anode, respectively[14,15]. Chen et al.[15] employed a sol-gel $Ti(Nb)O_x$ electron-extraction layer that was aimed to protect the perovskite against moisture from ambient atmosphere. Because of its poor electrical conductivity of $10^{-5}$ S cm$^{-1}$, the authors had to limit the thickness of the sol-gel $Ti(Nb)O_x$ to 10 nm. Moreover, it is known that sol-gel-derived layers state only relatively poor permeation barriers[16]. As a result, the device stability tests that afforded a lifetime of 1,000 h were done with additional encapsulation. Kaltenbrunner et al.[17] have used $Cr_2O_3$/Cr at the cathode side to protect the metal cathode and thereby to improve the stability under ambient conditions, but still the efficiency was found to drop to 80% of its initial value after only 28 h. Very recently, Guerrero et al.[18] confirmed the protective effect of the $Cr_2O_3$/Cr interlayer but still found some severe degradation after only a few hours. You et al.[14] have shown that devices based on indium tin oxide (ITO)/NiO/perovskite/ZnO/Al were stable for tens of days in ambient conditions, whereas a reference device based on ITO/PEDOT:PSS/perovskite/PCBM/Al showed severe degradation within half a day. Especially, the ZnO layer, which has been spin coated from a nanoparticle dispersion on top of the perovskite, has been claimed to protect the Al electrode against corrosion. Very recently, Bush et al.[19] have found that in their devices a similar nanoparticle-based ZnO layer next to the cathode did not provide any sufficient protection at elevated temperatures. As solar cells in outdoor conditions may reach a temperature significantly above 25 °C, the latter result appears to be extremely relevant. Other reports have evidenced rapid decomposition of $CH_3NH_3PbI_3$ when in direct contact with ZnO[20].

Here we propose an inverted cell architecture, where a bilayered aluminium-doped ZnO (AZO)/tin oxide ($SnO_x$) electron-extraction layer (EEL) affords perovskite solar cells with a remarkable resilience against moisture and heat. AZO is excellently suited to extract electrons from the lowest unoccupied molecular orbital of fullerenes[21]. However, the nanoparticle-based AZO layer does not provide sufficient protection against the ingress of moisture or the egress of perovskite decomposition products. Thus, we add a thin $SnO_x$ layer to form a bilayered AZO/$SnO_x$ EEL assembly. The $SnO_x$ is grown by low-temperature atomic layer deposition (ALD) at 80 °C that affords extremely dense, conformal and pinhole-free layers. Please note that there are reports where ALD layers have been directly deposited on top of perovskites to improve their resilience against ambient atmosphere and heat[22]. However, these ALD layers were not a functional part of a device, and therefore most of the experiments were based on electrical insulators like $Al_2O_3$. $SnO_x$ layers grown by ALD are optically highly transparent, electrically conductive and provide outstanding gas permeation barrier properties with a water vapour transmission rate as low as $7 \times 10^{-5}$ g (m$^{-2}$ day$^{-1}$), even when grown at a low temperature[23]. Their water vapour transmission rate is orders of magnitude better than that of sputtered ITO thin films or that of solution processed metal oxide layers. The electrical conductivity of the $SnO_x$ layer on the order of $5 \times 10^{-3}$ S cm$^{-1}$ allows us to place it between the sensitive electrode and the photoactive layers without adding any noticeable series resistance. Thereby, the $SnO_x$ provides outstanding protection of the perovskite against the ingress of moisture and, more importantly, at the same time it serves as permeation barrier against the out-diffusion of decomposition products of the perovskite. We will show that the sealing property of the $SnO_x$ layer contains the decomposition products inside the cell and, thereby, the overall decomposition of the perovskite is significantly suppressed. Although cells based on LiF/Al or AZO degrade within tens of hours in ambient air (23 °C and 50% relative humidity (rH)), devices based on AZO/$SnO_x$ show superior stability of their solar cell characteristics even on a timescale of >350 h. More strikingly, at 60 °C under inert atmosphere, AZO-based devices degrade within 100 h because of thermally induced decomposition of the perovskite. On the contrary, the suppressed out-diffusion of decomposition products because of the diffusion-barrier properties of the AZO/$SnO_x$ EEL affords cells that are essentially unchanged even after >1,000 h under the same conditions. Our paper reports a general strategy to achieve a substantially improved device lifetime in the case of photoactive materials that may come with concerns about their intrinsic compositional stability.

## Results

**Planar inverted cells.** The devices used in this study are based on a planar inverted cell layout, where the holes are extracted via the bottom electrode (substrate electrode). The layer sequence is shown in Fig. 1. As hole-extraction layer, PEDOT:PSS is used, whereas for electron extraction a 100 nm thick PCBM layer is deposited on top of the typically 180 nm thick $CH_3NH_3PbI_3$ perovskite layer. Directly adjacent to the PCBM we have used a 100 nm thick AZO layer prepared from a nanoparticle dispersion in isopropylalcohol. In some cells, a $SnO_x$ layer is deposited on top of the AZO layer by ALD at 80 °C to form the bilayered electron-extraction assembly (AZO/$SnO_x$). The nominal thickness of the $SnO_x$ layer is 20 nm. As the AZO layer is derived from a nanoparticle dispersion, it contains pores that are partially coated with $SnO_x$ because of the conformal nature of the ALD process. A more detailed discussion of the $SnO_x$ growth on top of the AZO layer is provided in the Supplementary Fig. 1. Please note that we have also prepared devices with $SnO_x$ deposited directly on top of PCBM, but the stability of the

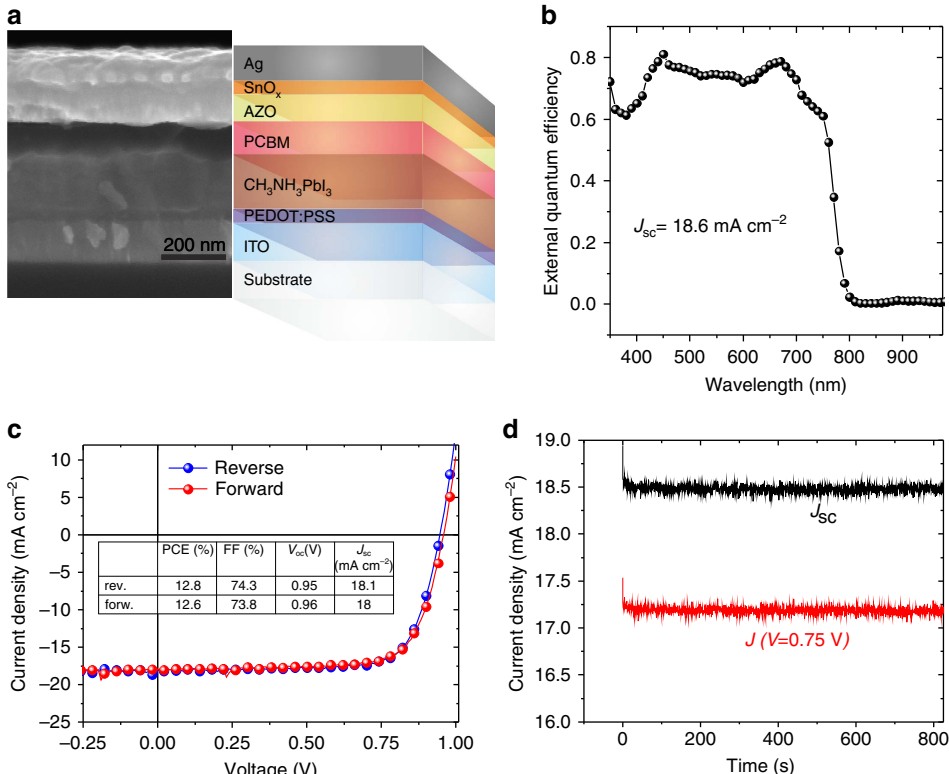

**Figure 1 | Basic device characteristics.** Scanning electron microscopy (SEM) image of the device cross-section along with the assignment of the respective layers (**a**). In some devices, the $SnO_x$ or the AZO layer has been omitted. Note that the contrast between the AZO and the $SnO_x$ layers in SEM is relatively poor. External quantum efficiency (EQE) of a representative device based on AZO/$SnO_x$ (**b**). J/V characteristics measured in forward and reverse direction (**c**), and current density versus time under 0 V (black) and 0.75 V (red) bias (**d**).

resulting devices was inferior to that of the AZO/$SnO_x$ cells (see discussion below). Reference devices based on ITO/PEDOT:PSS/Perovskite/PCBM/LiF/Al have also been included in our study.

The external quantum efficiency (EQE) spectrum of a typical AZO/$SnO_x$-based perovskite solar cell is shown in Fig. 1b. The J/V characteristics of the device measured in forward and reverse direction (scan speed: $500\,mV\,s^{-1}$) are presented in Fig. 1c. No substantial hysteresis is found in the J/V data. The extracted characteristics $V_{oc} = 0.95\,V$, FF = 74% and a $J_{sc} = 18\,mA\,cm^{-2}$ result in a PCE of 12.8%. We want to note that the perovskite precursor used in this study was commercially obtained from Ossila Ltd with a typically specified efficiency in the range of 11–13% (see Methods for details).

Significantly higher PCEs can be achieved by using, for example, optimized precursor inks or mixed-cation/mixed-halide perovskite absorbers[24]. As the EEL is in the core of this report, further optimization of the perovskite system itself was considered to be beyond the scope here.

**Stability in ambient air.** A very striking advantage of the bilayered AZO/$SnO_x$ EEL in direct comparison with AZO is displayed in Fig. 2. Here, the stability of the cell characteristics is shown for devices that were continuously exposed to ambient air at 23 °C and 50% relative humidity. The devices that were based on a single layer of AZO as EEL degrade swiftly within ∼ 50 h. A severe decay of the FF from ∼ 70% to < 25% and a concomitant drop of the $J_{sc}$ from ∼ 17 to 8.2 mA cm$^{-2}$ is found. Opposed to that, the characteristics of the corresponding cells with the bilayered EEL of AZO/$SnO_x$ do not show any

degradation even after > 300 h in air. Note that the thickness of the $SnO_x$ layer is only 20 nm in these devices. There is a slight increase of the $V_{oc}$ in the AZO/$SnO_x$ samples that happens on a timescale of several tens of hours. This increase is only observed in the AZO/$SnO_x$ samples, as the LiF/Al and the AZO cells already undergo strong degradation on the same timescale. A similar increase of $V_{oc}$ on such a long timescale has been seen by other authors and it has been explained by an ageing effect that reduces the density of trap states in the perovskite[25]. As a reference, the cells based on LiF/Al are essentially degraded within ∼ 1 day.

In order to analyse the reasons underlying the strikingly different degradation phenomena shown in Fig. 2, we have performed X-ray diffraction (XRD) and X-ray photoelectron spectroscopy (XPS) studies of the respective device structures. Note that in case of the XPS measurements, a thinner top Ag electrode was chosen (10 nm) compared with the actual solar cells (100 nm) because of the small probing depth of the technique. Looking first at the AZO-based device, the XRD spectrum shows the (110) and (220) reflections of tetragonal MAPbI₃ at angles of 14.11° and 28.14, respectively. As this XRD measurement has been made on a full device stack, further signals related to the Ag and ITO electrodes are visible. A detailed assignment of these peaks is shown in the Supplementary Fig. 3. Only a very weak signal due to $PbI_2$ was found that did not significantly increase even after 7 days in air (Fig. 3a). Thus, no severe decomposition of the perovskite to $PbI_2$ upon storage in air can be inferred from the XRD data. On the other hand, when XPS was used to look for degradation products on the surface, a notable amount of iodine was evidenced on the AZO sample (Fig. 3b) after exposure to air for 2 days, whereas on identical samples kept in nitrogen no such

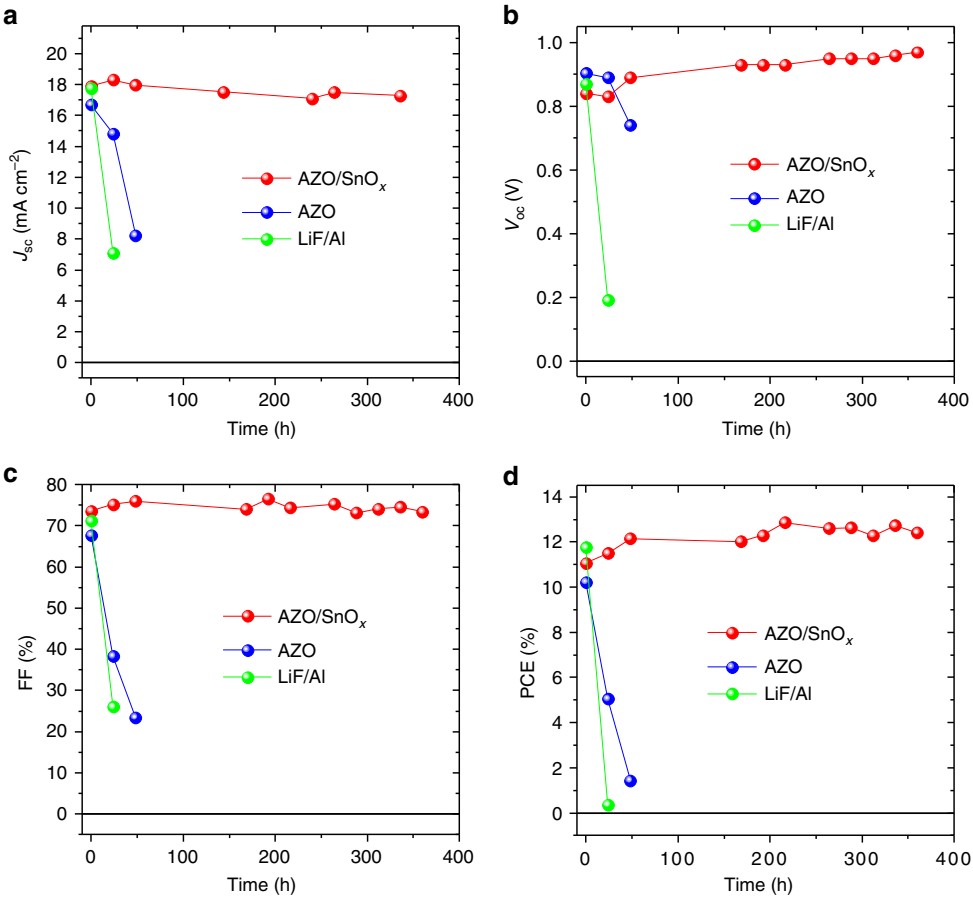

**Figure 2 | Stability upon storage in ambient air.** Characteristics of inverted perovskite collar cells with varied cathode/electron extraction assemblies (AZO/SnO$_x$/Ag, AZO/Ag and LiF/Al) versus time of storage in ambient air (at 23 °C and 50% rH). $J_{sc}$ (**a**), $V_{oc}$ (**b**), FF (**c**) and PCE (**d**). The characteristics have been determined from the J/V measurements in reverse direction (examples in Supplementary Fig. 2).

iodine signal is found. Please note that the peak of ∼ 622 eV in the XPS spectrum of the aged AZO cell does not represent another bonding state of the iodine, but it is rather a so-called shake-up peak. Such shake-up peaks are commonly observed in strong XPS signals. The survey XPS spectra can be found in the Supplementary Fig. 4. Plan-view scanning electron microscopy (SEM) images of the AZO samples aged in air show the formation of spot-like degradation motifs (Fig. 3c,d). These spots point to the local ingress of moisture and out-diffusion of decomposition products of the perovskite, supported by the appearance of the iodine signal in XPS. In the degraded spots, needle-like features occur. Similar features have been reported for Ag electrodes corroded by the decomposition products of MAPbI$_3$ (ref. 26). Our findings are also in line with the results of Kato et al.[10] who unveiled a moisture promoted corrosion mechanism of Ag which resulted in the formation of products like AgI. The role of water in the electrode corrosion is further clarified below in cells aged under inert conditions.

To explain the spot-like degradation, we can assume that structural defects, such as pinholes, in the AZO NP layer would give rise to locally enhanced water permeation. Opposed to that, the sample with the bilayered EEL of AZO/SnO$_x$ does not show any sign of degradation in SEM (Fig. 3e) under identical conditions. In XPS, only a negligible amount of iodine at the surface of the AZO/SnO$_x$ samples can be detected. For comparison, we have also studied layer stacks where only SnO$_x$, that is, without AZO between the PCBM and the SnO$_x$, has been used as EEL. We have recently shown that SnO$_x$ with

a work function of 4.1 eV forms an excellent EEL layer in inverted organic solar cells[27]. Here, the resulting perovskite solar cells based on SnO$_x$ were more stable than the devices based on AZO. However, they showed a statistical failure on a timescale of 100 h (Supplementary Fig. 5). This finding could be explained by a nonideal ALD growth of the SnO$_x$ directly on top of the organic PCBM layer. ALD as a chemical deposition technique relies on self-limiting chemical reactions on the sample surface[28]. For the ALD deposition of oxide layers, –OH surface groups form the nucleation sites for layer growth. In this sense, on organic surfaces like PCBM the nucleation of an ALD layer may deviate significantly from the growth on a surface rich of –OH groups[29]. Therefore, we can assume that the SnO$_x$ layer grown on PCBM is not free of some pin-hole defects that compromise its functionality as diffusion barrier. Consequently, substantially more iodine is detected on the surface after exposure to air compared with the AZO/SnO$_x$ device (Fig. 3b). Thus, we conclude that the AZO serves two purposes in the bilayered AZO/SnO$_x$ EEL: (1) its electronic structure has been shown to facilitate electron extraction from the PCBM (ref. 21) and (2) the AZO layer provides an improved (oxide-) surface for the nucleation of the subsequent SnO$_x$ ALD layer that can thus from a dense, pin-hole-free permeation barrier. Figure 3f,g shows schemes portraying the specific differences between AZO and AZO/SnO$_x$ as EEL in devices exposed to air.

As indicated before, the AZO layer prepared from the NP dispersion does not provide a significant diffusion barrier against the local ingress of moisture from the ambient air.

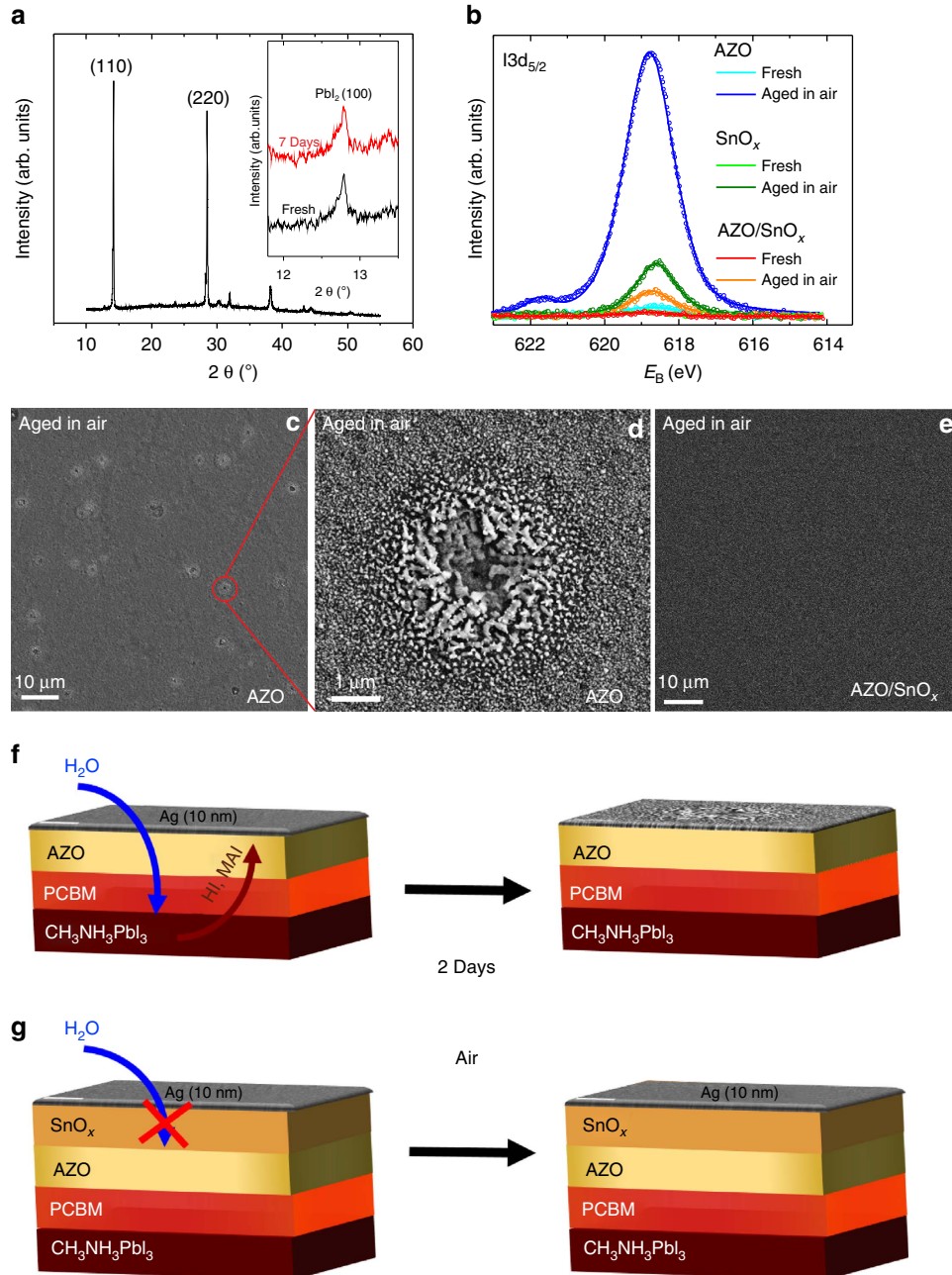

**Figure 3 | Unravelling the ageing mechanism in air.** XRD spectrum of a perovskite cell based on AZO as EEL after storage in air for 7 days (inset: magnified view of the (100) $PbI_2$ reflection for a fresh sample and one that has been stored in air for 7 days; spectra were vertically offset for clarity) (**a**). Photoemission spectra of the iodine $I3d_{5/2}$ peak for fresh and aged cells based on AZO, $SnO_x$ and bilayered AZO/$SnO_x$ EELs, respectively (**b**). The aged samples were stored in air for 2 days. Corresponding plan-view SEM images of the 10 nm Ag layer in case of the AZO sample (**c,d**), and for the AZO/$SnO_x$ sample (**e**). Schemes of ageing for the AZO and the AZO/$SnO_x$ samples (**f,g**).

As a result, decomposition of the $MAPbI_3$ perovskite occurs and yields iodine-based products that migrate to the surface. In the case of AZO/$SnO_x$, substantially better moisture barrier properties prevent the penetration of water. The claim of superior moisture barrier properties of the bilayered AZO/$SnO_x$ EEL compared with that of the AZO NP layer is also supported by Ca-corrosion tests (Supplementary Fig. 6).

It has to be noted that the detrimental effects of ambient air could in principle be avoided by using a proper encapsulation of the entire solar cell. However, the out-diffusion of decomposition products from the perovskite and the degradation of the metal electrode (Fig. 3c,d) may impose a more fundamental

issue. This is even more important, as earlier reports evidenced the thermally activated decomposition of the $MAPbI_3$ perovskite even under inert atmosphere[7].

**Stability at elevated temperatures.** To study the long-term stability of our solar cells under inert conditions, we have stored a set of devices in a glove box under $N_2$ atmosphere. All the devices were placed on a hot plate at 60 °C and their characteristics were measured repeatedly. Most remarkably, the characteristics of devices based on AZO EELs degraded on a timescale of 100 h (Fig. 4). Although the $J_{sc}$ and $V_{oc}$ of these devices remain

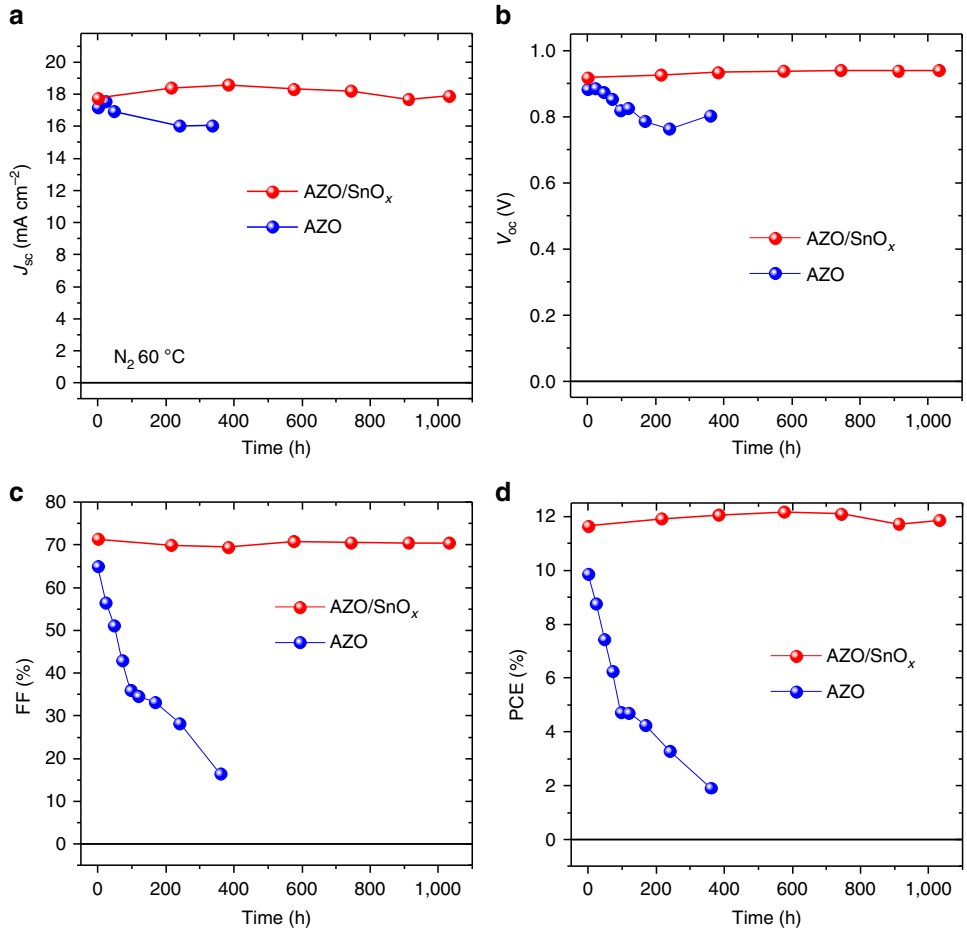

**Figure 4 | Stability upon storage at 60 °C in nitrogen atmosphere.** Characteristics of inverted perovskite cells with varied cathode electron-extraction assemblies (AZO/SnO$_x$/Ag, AZO/Ag) versus time. $J_{sc}$ (**a**), $V_{oc}$ (**b**), FF (**c**) and PCE (**d**). In this set of samples, the nominal thickness of the ALD SnO$_x$ layer was 20 nm and that of the Ag electrode was 100 nm. The characteristics have been determined from the $J/V$ measurements in reverse direction (Supplementary Fig. 7).

fairly stable, the decay of the FF results in the severe drop of PCE to ~2% after 360 h. The degradation motif in the AZO-based cells is essentially different from that found in ambient air, where all cell characteristics decayed within a day (Fig. 2). Again in striking contrast, the characteristics of devices based on the bilayered AZO/SnO$_x$ EEL remain almost unchanged even after 1,032 h. Note that the hysteresis did not increase in the course of ageing.

To unravel the mechanisms behind these results, we have again performed XRD and XPS studies of the respective device structures. After heating the device stacks for 6 days under inert atmosphere, the XRD spectrum of the AZO samples reveals a significant signal due to PbI$_2$ that is absent in pristine stacks (Fig. 5a). The formation of PbI$_2$ indicates the thermally activated decomposition of the perovskite upon heating, in agreement with earlier reports[7]. Opposed to that, the devices based on the bilayered AZO/SnO$_x$ EEL do not show any signature of a PbI$_2$ phase after identical ageing conditions (Fig. 5b).

The formation of PbI$_2$ in the AZO samples is in contrast to the degradation experiments in air, where no PbI$_2$ phase in the AZO samples was found in XRD (Fig. 3a). At the same time, the I3d$_{5/2}$ XPS spectra (Fig. 5c) do not reveal any significant increase of the amount of iodine at the surface of any of the samples aged in N$_2$. Please note that the noise in the XPS spectra indicates that the concentration of I3d$_{5/2}$ at the surface of all the samples is close to the detection limit and variations between the

samples are within the measurement error. A plan-view SEM image of the aged AZO samples does not reveal any degradation of the Ag electrode, in contrast to the experiment where the ageing took place in air (Fig. 3c,d). Thus, we conclude that in the case of AZO, volatile decomposition products of the MAPbI$_3$ perovskite evaporate from the cell stack without degradation of the Ag electrode. In air, other than in N$_2$, the presence of water would promote the corrosion of Ag because of CH$_3$NH$_3$I and other halide compounds[10]. However, the loss of CH$_3$NH$_3$I etc. drives the further decomposition of the perovskite.

## Discussion

Taken together, the device characteristics (Fig. 4) and the XRD and XPS spectroscopy point to a scheme of degradation as shown in Fig. 5e,f. Under inert atmosphere, the thermally activated degradation of the perovskite leads to decomposition products that can easily diffuse through the PCBM/AZO/Ag layers on top of the perovskite. The less volatile PbI$_2$ phase remains in the active layer leading to a substantial decay of the FF of the corresponding devices (Fig. 4c). Moreover, it has been shown earlier that a change in film stoichiometry (Pb poor to Pb rich) may substantially alter the electronic properties of the perovskite and especially the position of the conduction and valence band with respect to the vacuum level[30]. Towards

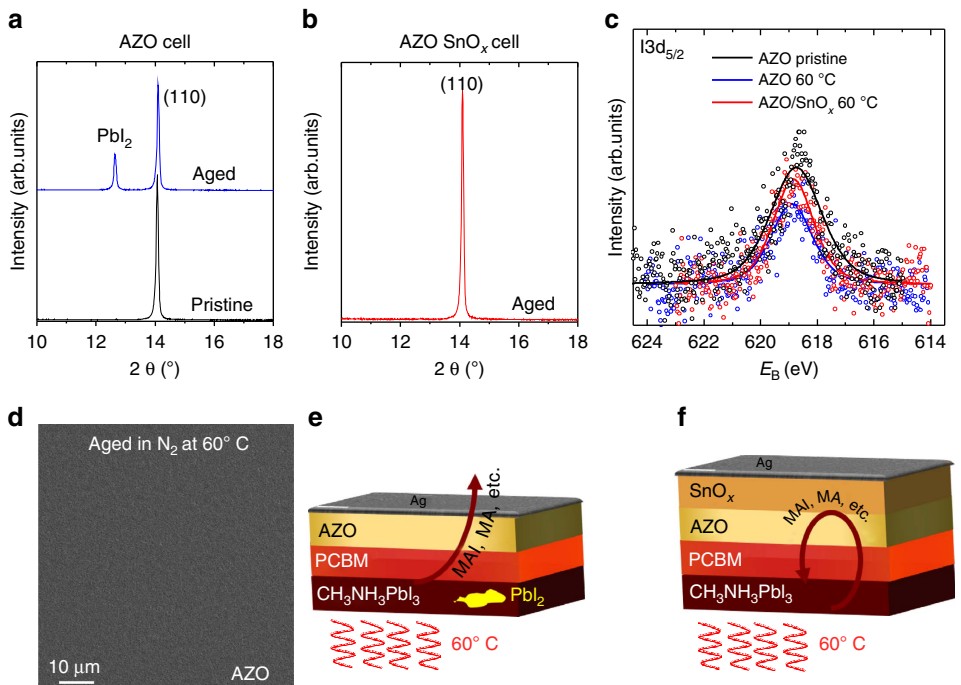

**Figure 5 | Why impermeable AZO/SnO$_x$ EELs afford stable devices.** XRD spectra of a perovskite cell based on AZO and AZO/SnO$_x$ as EEL before and after heating in nitrogen for 6 days. (**a,b**) Photoemission spectra of the iodine I3d$_{5/2}$ peak for cells based on AZO and bilayered AZO/SnO$_x$ EELs, respectively. (**c**) As a reference, an AZO sample stored in N$_2$ ambient has been used. The other measurements are for samples aged at 60 °C in nitrogen for 6 days. The open symbols represent the raw measurement data, whereas the solid lines are a result of a peak fit. The colours are defined in the figure legend. Note that the top Ag electrode was chosen to be 10 nm. Corresponding plan-view SEM image of the 10 nm Ag layer in case of the aged AZO sample. (**d**) Schemes of ageing and decomposition of the perovskite in case of the AZO (**e**) and the AZO/SnO$_x$ samples (**f**). The yellow spot in the scheme of the AZO sample symbolizes the formation of PbI$_2$.

increasingly Pb-rich conditions, the ionization energy of the perovskite has been shown to increase significantly to values in excess of 6 eV that would render the extraction of holes more and more challenging.

Opposed to the case of the AZO cells, the AZO/SnO$_x$-based cells do not show any formation of PbI$_2$ under otherwise identical conditions. We therefore conclude that the decomposition products are contained inside the cell stack because of the dense SnO$_x$ layer forming a permeation barrier. Thereby, the equilibrium of the decomposition/formation reaction would be shifted to stabilize the perovskite. Thus, the decomposition of the MAPbI$_3$ obviously is significantly slowed down, and the cells do not show significant degradation even after heating for >1,000 h (Fig. 4). It has to be noted that it is favourable to position the permeation barrier between the perovskite and the Ag electrode, as thereby the known long-term corrosive effect of residual halide components on the Ag electrode[31] and on the interface EEL/Ag can be efficiently suppressed. In further work, this concept could even be extended to a setup, in which the perovskite layer is sandwiched between diffusion barriers at the cathode and anode side, in case permeable substrates like polymer foils are used. For example, there are reports on using ALD to prepare V$_2$O$_5$ layers[32]. V$_2$O$_5$ is a high work function transition metal oxide that has successfully been used for hole extraction in organic and perovskite solar cells[33–36]. ALD prepared V$_2$O$_5$ would presumably provide similar gas permeation barrier properties as those shown for the SnO$_x$. Thereby, the intrinsic decomposition of the perovskite active layers would be substantially delayed or even prevented as volatile decomposition products would be hindered from evaporating.

Aside from the so-called mono-stress conditions like elevated temperature, multi-stressing can be considered. To this end, we conducted set of long-term light soaking tests at 60 °C under N$_2$, in which the devices were simultaneously illuminated with a white light-emitting diode (LED) to achieve the same $J_{sc}$ as upon AM1.5G solar irradiation (for details see the Methods section). The results of this multi-stress experiment are shown in the Supplementary Fig. 8. Briefly, again the AZO/SnO$_x$ cells are substantially more stable than the AZO cells under concomitant heat and illumination. This difference is in part because of the suppressed decomposition of the perovskite owing to heat in case of the AZO/SnO$_x$ (in agreement with the discussion of Fig. 4). However, there is a clear degradation even of the AZO/SnO$_x$ samples where the PCE decays to roughly 60% of its initial value after 300 h. This is in contrast to the case where only thermal stress has been used. Earlier work has unravelled the photo-induced degradation of solar cells based on MAPbI$_3$. Whereas Misra et al.[37] identified a photoactivated decomposition mechanism, more recently Nie et al.[38] reported that continuous illumination of MAPbI$_3$ caused the formation of trap states that spoiled the solar cell performance, especially $J_{sc}$. Although our sealing approach based on the impermeable AZO/SnO$_x$ electron extraction layer has been shown to efficiently suppress the decomposition of the perovskite, it cannot suppress the formation of trap states in the perovskite. Similar to the report of Nie et al.[38] the cell characteristics recover after stressing, if the cells are kept in darkness (Supplementary Fig. 8f). This is why we conclude that photo-induced trap-state formation occurs in our cells. However, there have been recent reports confirming that photo-induced degradation is not a general problem of organolead halide perovskites, and some optimized mixed cation/

mixed halide perovskite active materials were shown to be far less prone to light soaking degradation[24,39]. Heat was found to be a dominating source of degradation in these mixed cation perovskite cells[40]. Note that our inverted device structure based on the impermeable AZO/SnO$_x$ electron extraction layer is generally applicable and can also accommodate these next-generation perovskite photoactive systems with enhanced stability and efficiency[41].

In summary, we have shown an ultra-stable inverted cell architecture, where a bilayered AZO/SnO$_x$ EEL has been demonstrated to afford perovskite solar cells with a remarkable resilience. The 20 nm thin ALD grown SnO$_x$ has been evidenced to form an outstandingly dense gas permeation barrier that effectively hinders the ingress of moisture towards the perovskite and—more importantly—it prevents the egress of decomposition products like CH$_3$NH$_3$I, HI out of the perovskite.

Although the efficiency of cells based on LiF/Al or AZO degraded to <50% of the initial value within less than a day in ambient air, devices based on AZO/SnO$_x$ showed superior stability of their solar cell characteristics even on a timescale of >350 h. Under inert atmosphere, AZO-based devices degraded to roughly 50% of its initial efficiency within 100 h because of thermally induced decomposition of the perovskite. On the contrary, the suppressed out-diffusion of decomposition products because of the diffusion-barrier properties of the AZO/SnO$_x$ EEL has been shown to significantly slow down the thermally activated decomposition of the perovskite. This afforded cells that did not change even after >1,000 h. The concept of impermeable EELs is generally applicable and therefore is expected to provide an avenue to achieve a substantially improved device lifetime of solar cells based on organometal halide perovskites even beyond MAPbI$_3$.

## Methods

**Material synthesis and device preparation.** The inverted perovskite solar cells studied in this work are based on the following layer sequence: glass/ITO/PEDOT:PSS/perovskite/PCBM/EEL/Ag (see Fig. 1a). The PEDOT layer (AI4083) has been spin coated in ambient air and dried on a hot plate at 120 °C for 20 min in air and for another 20 min in a glove box under N$_2$ atmosphere. The perovskite layer has been spin coated under inert atmosphere from a commercially available precursor solution (Ossila) and the resulting layers were thermally annealed at 100 °C for 80 min. The typical layer thickness was 180 nm.

On top of the perovskite, PCBM (American Dye Sorce Inc., ADS61BFA) was spin coated from a solution in chlorobenzene (concentration: 100 mg ml$^{-1}$). AZO has been deposited from a commercial nanoparticle dispersion (2.5 wt% in isopropanol, Prod. No. 8045, Nanograde AG, Switzerland). Tin oxide has been prepared by atomic layer deposition in a Beneq TFS 200 system (base pressure 1.5 mbar). As precursors, tetrakis(dimethylamino)tin(IV) (TDMASn), kept at 45 °C and water, kept at room temperature, were used. At a substrate temperature of 80 °C the growth rate per cycle was 1.1 Å. Further details on the materials properties of the SnO$_x$ can be found in our earlier work[23,27]. Ag (100 or 10 nm) layers were thermally evaporated in high vacuum (10$^{-7}$ mbar). In some reference devices, LiF (1 nm) and Al (100 nm) have been thermally evaporated as cathode.

**Materials and device characterization.** The electrical conductivity was measured with the Van der Pauw method.

The SEM studies were conducted using Neon 40 (Zeiss).

XRD was measured using a monochromatic Cu-Kα$_{1,2}$ source (Philips X'Pert Pro MPD). XPS has been performed using a non-monochromatic Mg K$_\alpha$ X-ray source (VG) and a Phoibos 100 (Specs) electron analyser.

Rutherford backscattering was measured at RUBION (University of Bochum, Germany) using a dynamitron tandem accelerator with 2 MeV $^4$He$^+$ beam (beam current of 20–40 nA). A silicon surface barrier detector was placed at an angle of 160° with respect to the beam axis. The solid angle of the detector was 1.91 msrad.

The solar cells were characterized in ambient air without encapsulation using a Keithley 2400-C source meter and a solar simulator (300 W Newport, model 91160, AM1.5G, 100 mW cm$^{-2}$). We used a commercial Si reference cell (Rera Systems) that is IEC 60904–9 compliant and certified. Short circuit current was obtained from the EQE data. EQE has been measured with a home-built tunable light source, calibrated with a power meter (Thorlabs). The spectral mismatch factor was 1.163.

Some of the devices have been kept in a glovebox under N$_2$ atmosphere on a hot plate at 60 °C. The J/V characteristics were measured in forward and reverse direction at a scanning speed of 500 mV s$^{-1}$. Cells were measured using a mask with round aperture of 1.5 mm diameter. Stabilized current was measured in the same J/V setup using a voltage source.

Long-term light soaking tests at 60 °C under N$_2$ were conducted by simultaneously illuminating the cells with a white LED to achieve the same J$_{sc}$ as upon AM1.5G solar irradiation. The LED spectrum is shown in the Supplementary Fig. 8e. The cells were mounted in a home-built oven inside a glove box. They were kept in short circuit conditions during the ageing experiment.

**Data availability.** The data that support the findings of this study are available from the corresponding author on request.

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

## Acknowledgements

We acknowledge the German Federal Ministry for Education and Research (Grant No. 03EK3529E) and the Deutsche Forschungsgemeinschaft (DFG) (Grants: RI1551/4-2) for financial support. We acknowledge the Ministry of Science of the state of NRW for funding within the PeroBOOST (EFRE) project. The research leading to these results has received partial funding from the European Unions 7th Framework Programme under Grant Agreement no. 604148 (MUJULIMA).

## Author contributions

T.R. and K.O.B. conceived and designed the experiments. S.O. and K.M. contributed the XPS and plan-view SEM analysis. N.P. did the XRD study. D.R. contributed the Rutherford backscattering (RBS) analysis. K.O.B., T.B., T.H. and J.Z. performed the experimental work on the perovskite cell devices. L.H. and T.G. contributed the tin oxide ALD layers as well as electrical and permeation barrier studies. R.H. did the cross-section SEM measurements. M.F.O. and N.A.L. provided the AZO dispersions along with the expertise of their processing. All authors discussed the results and were involved in the writing.

## Additional information

**Competing financial interests:** The authors declare no competing financial interests.

**Publisher's note**: 

