## [Peer Review File · Nature Communications]

Reviewers' comments:

Reviewer #1 (Remarks to the Author):

This manuscript describes an inverted perovskite photovoltaic in which the electron extraction layer consists of PCBM-only, PCBM + AZO nanoparticles, or PCBM + AZO nanoparticles + SnO₂ ALD overcoat. The work convincingly shows an improvement in PV stability with the 3-step overcoat as well as supports a hypothesis for the mechanism by which the devices are stabilized.

The originality and interest of the report is good but modest. Although the authors do not highlight previous ALD-over-perovskite work there are other examples including *J. Mater. Chem. A*, 2015, 3, 20092-20096.

The thickness of the SnO₂ layer is suggested to be 20 nm, but this ignores the high likelihood that the ALD SnO_x has coated a significant fraction of porous interior of the nanoparticle AZO on which it is grown.

Overall the paper is modestly well written and its conclusions are sound. However, the impact appears to be below the very high bar of nature communications as the lifetime is good (350 hrs) but never pushed to the real boundaries previous perovskite systems (1000 hrs of illumination, high relative temperature and humidity, etc) the methods is not totally novel, and efficiency is modest (although unoptimized). While I don't believe all high impact papers need to be 18+ % efficient, the mechanism by which the most efficient devices fail is important.

Suggested improvements are stability tests that push the device to failure including light soaking under temperature and humidity followed by publication in a journal similar to *E&ES*.

Reviewer #2 (Remarks to the Author):

This manuscript described the use of a bi-layer oxide AZO/SnO_x as electron extraction layer to improve the stability of perovskite solar cell. Stability is a crucial issue for perovskite solar cell before it can be considered as a promising alternative for current existing technologies. The finding and phenomena reported in this communication could be interesting for the community. However, several issues need to be addressed before the acceptance by high-ranking journal.

1. The results of the most successful device on stability [Ref 17, *Science* 350, 944-948 (2015)] shall be briefly described and discussed in the introduction. The device architecture of this communication is conceptually similar to that *Science* paper, except the use of bilayer oxide. I think the advantages and novelty (vs that *Science* paper work) shall be clearly expressed in the introduction to show the importance of this work. Otherwise, changing materials can only be considered as a minor progression or follow-up.
2. For the XRD results, the spectra seem to contain only the diffraction pattern of perovskite. As the author said in the manuscript the sample is a full device, I think the author shall see the diffraction of those constituent layers to prove that the sample is the full device and cover with EEL and metal contact. Please clarify this issue. It is quite unusual that under ambient condition, no PbI₂ signal were observed. In general the formation of PbI₂ prevails extremely fast in air. The explanation of "non-crystalline nature of the degradation product" is not very convincing as no amorphous feature in the

XRD spectra.

3. For the XPS results, the baseline of fresh devices made with SnOx and AZO/SnOx shall be included. Since XPS is a surface technique that investigates on the first few 3~5 nm, I wonder if the perovskite can be detected if protected after 10 nm Ag and AZO. The authors shall provide the whole spectrum to show what they have detected on the surface. The signal of Eb close to 622 eV for the blue line on Fig 3 (b) shall be identified and discussed. Is it something other than AgI?

4. Are the SEM images taken on top of Ag or on top of oxides? I think that matters for proposing the mechanism.

5. To demonstrate real stability for solar cells, light soaking is necessary. Especially for perovskite solar cell, light soaking results can really substantiate the claim for long term stability. To be published in such high impact journal like Nature Communications, I think the authors shall show stability under light soaking.

6. The unit of EQE on Figure 1b is wrong.

7. The first inverted perovskite solar cell using PEDOT:PSS and PCBM shall be cited in the introduction. Adv. Mater. 2013, 25, 3727.

Overall, I suggest a minor revision for this manuscript.

Reviewer #3 (Remarks to the Author):

The manuscript by Brinkmann et al address the use of AZO/SnOx electron extractor layers in hybrid perovskite solar cells. More specifically, the paper reports improved device stability when such AZO/SnOx layers are used in inverted type solar cells. The paper is interesting but in the current state the paper is not suitable for a journal like Nature Communications. There are a number of significant issues that must be addressed.

1 The more detailed mechanism of how the combined AZO/SnOx layer interlayer improves stability needs to be established. This is not at all clear from the data presented in the paper.

2 Following from above, what is the mechanism by which the SnOx layer improves stability? How does it block water getting in the device? Surely most of the diffusion occurs from the sides of the devices and through the silver? Why can't the metal electrode act as a barrier to water? It is not beyond the realms of possibility that approximately 100 nm thick (or several 10's nm thick) metal electrodes can function good physical blocking layers. Perhaps this should be investigated.

3 The authors suggest that the silver electrode may be involved in the degradation of the device in presence of water. Have the authors tried to replace the silver electrode with lets say gold (which is known to be more stable than silver) in control devices that do not contain the SnOx layer: ITO/PEDOT/active layer/gold.

5 The authors should provide more information about how the device stability experiments were performed. Was this done with maximum power point tracking? Were the experiments performed whilst continuously illuminating the solar cells under load with the solar cells put in ambient environment? Or, are the authors just referring "shelf life" stability. To get a true indication of stability it is important that the authors consider the former (e.g) measurements under continuous illumination under load with MPP tracking.

6 In the temperature stability measurements shown in figure 4. It is shown the drop in performance is mainly due to a drop in the fill factor. The authors should discuss the origin of this. The authors should also perhaps repeat temperature stability experiments using a thicker electrode layer - certainly

thicker than 10nm !! It is not surprising that with such thin layers of silver (e.g. 10nm) increase the chance that the degradation products escape.

REVIEWERS' COMMENTS:

Reviewer #1 (Remarks to the Author):

I am satisfied with the revisions to the manuscript. Although a simultaneous temperature and moisture (damp heat) test, which all commercial PV must pass, would be most interesting there is sufficient data included here for an initial report. The changes raise the quality of this revised work to those required by Nature Comm. I support prompt publication.

Reviewer #2 (Remarks to the Author):

The revised version has addressed most of the comments and questions raised by the reviewer properly. I would recommend its publication.

Reviewer #3 (Remarks to the Author):

Based on the responses and reading the revised paper the referee believes that the paper may now be suitable for publication in Nature communications.

Point 1 and 2: The role of metal contact versus SnOx layer in protecting against water vapour ingress has been clarified. The comparison between vapour transmission rates for metal versus ALD grown SnOx is welcome.

Point 3: Though a comparison of gold and silver as an electrode and the investigation of the use of SnOx/AZO as a blocking layer (ie reducing migration of metal atoms from electrode) are important issues the referee accepts that this is potentially beyond the scope of the current work.

Point 5: The additional solar cell stability experiments (Figure S8) are welcome and serve to clarify the concerns of referee 3.

Point 6: comments regarding thickness of Ag layer clarified. Concerning the origin of the drop in the fill factor (FF): the discussion is welcome. However, the referee has an additional question here: Did the authors observe a change in hysteresis in fresh versus aged samples. (referring to a device that is treated in the same way as in Figure 5; main manuscript). For example, a device that is using an AZO/SnOx extraction layer (with and without heating at 60C)?

Reviewer #1 (Remarks to the Author):

This manuscript describes an inverted perovskite photovoltaic in which the electron extraction layer consists of PCBM-only, PCBM + AZO nanoparticles, or PCBM + AZO nanoparticles + SnO₂ ALD overcoat. The work convincingly shows an improvement in PV stability with the 3-step overcoat as well as supports a hypothesis for the mechanism by which the devices are stabilized.

Reply:

We appreciate the referee's assessment of our work. We were pleased to read that he finds our results convincing.

The originality and interest of the report is good but modest. Although the authors do not highlight previous ALD-over-perovskite work there are other examples including J. Mater. Chem. A, 2015,3, 20092-20096.

Reply:

We are aware of the previous work, in which single ALD layers have been deposited on top of a perovskite to protect it against ambient gases (e.g. moisture) and thereby to improve its resilience in ambient air. The functionality of ALD layers in these earlier papers is that of a permeation barrier. These ALD layers are not a functional part of a device structure and their electrical properties are more or less irrelevant and neglected.

*Opposed to that, our work is essentially different, as the SnO_x ALD layer is a functional part of our devices. The SnO_x layer is unique in a sense that it provides outstanding gas permeation barrier properties with a water vapor transmission rate orders of magnitude better than that of sputtered ITO or that of solution processed metal-oxides, and at the same time it is optically highly transparent **and** electrically conductive. Thus it can be positioned right between the sensitive metal cathode and the photo-active perovskite. Thereby, the SnO_x layer provides outstanding protection of the perovskite against the ingress of moisture and, even more importantly, at the same time it serves as powerful permeation barrier against the out-diffusion of decomposition products of the perovskite, e.g. CH₃NH₃I, (CH₃)₃N, CH₃I, HI, etc. We show that the sealing properties of the SnO_x layer contain the decomposition products inside the cell, and thereby, the overall decomposition of the perovskite is significantly suppressed. We feel that this concept goes significantly beyond the "ALD-on-top-of-perovskite" reports cited by the referee.*

To make things more clear we referenced the above citation and added a sentence to clarify the differences:

"Please note, there are reports where ALD layers have been directly deposited on top of a perovskite layer to improve its resilience against ambient atmosphere and heat.¹ However,

these ALD layers were not a functional part of a device, and therefore most of the experiments were based on electrical insulators like Al₂O₃.”

The thickness of the SnO₂ layer is suggested to be 20 nm, but this ignores the high likelihood that the ALD SnO_x has coated a significant fraction of porous interior of the nanoparticle AZO on which it is grown.

Reply:

*This is an excellent point! Indeed, ALD is known for its conformal, shadow-free type of growth which implies the coating of the interior of porous media. With the specification of a 20 nm thick SnO_x layer we referred to a **nominal** thickness as derived when the layer is coated onto a flat, non-porous substrate. We believe that the ability of the SnO_x ALD layer to seal pinholes and/or pores will substantially improve the permeation barrier properties of the AZO/SnO_x assembly vs. that of a single AZO layer.*

*To analyze this point deeper and to address the point raised by the referee, we performed Rutherford backscattering (RBS) measurements on two samples as shown in the Figure below (**which became the new Figure S1 in the revised supporting information**). Specifically, we have deposited SnO_x in one process on top of an AZO layer grown by ALD (ALD-AZO), which is thereby expected to have no pinholes, and on top of an AZO layer derived from a NP dispersion – both AZO layers have an identical thickness. From the RBS spectra attributed to Sn, we are able to determine the number of Sn atoms per area, which is indicative of the amount of SnO_x actually deposited. For the SnO_x on top of the pin-hole-free ALD-AZO we measured $(44.5 \pm 0.2) \times 10^{15}$ Sn-atoms/cm², which is in excellent agreement with our earlier RBS measurements of ALD grown SnO_x layers (on top of a flat Silicon wafer) with a thickness of 20 nm (determined by stylus profilometry). On the contrary, for the SnO_x deposited on top of the AZO (NP) layer we found $(72.2 \pm 0.9) \times 10^{15}$ Sn-atoms/cm², which is a factor of 1.62 higher than for the non-porous substrate. This result clearly supports the hypothesis of the ALD growth of the SnO_x inside the pores of the nanoparticle derived AZO layer, which increases the total amount of SnO_x deposited.*

To clarify the growth of the SnO_x on top of the AZO and into the pores, we added in the revised version of the manuscript:

“The nominal thickness of the SnO_x layer is 20 nm. As the AZO layer is derived from a nanoparticle dispersion, it contains pores which are partially coated with SnO_x due to the conformal nature of the ALD process. A more detailed discussion of the SnO_x growth on top of the AZO layer is provided in the supporting information.”

Figure S1: Layer sequence of the samples used to study the ALD growth of the SnO_x on top of an AZO layer grown by ALD, which is expected to have no pinholes (a), and on top of an AZO layer derived from a nanoparticle dispersion (b). The resulting RBS spectra attributed to Sn, which allow to determine the area density of Sn atoms (c).

We also added to the experimental section: "Rutherford backscattering (RBS) was measured at the RUBION (University of Bochum, Germany), using a dynamitron tandem accelerator with 2 MeV ⁴He⁺ beam (beam current of 20-40 nA). A silicon surface barrier detector was placed at an angle of 160 degree with respect to the beam axis. The solid angle of the detector was 1.91 msrad."

Overall the paper is modestly well written and its conclusions are sound. However, the impact appears to be below the very high bar of nature communications as the lifetime is good (350 hrs) but never pushed to the real boundaries previous pervoskite systems (1000 hrs of illumination, high relative temperature and humidity, etc) the methods is not totally novel, and efficiency is modest (although unoptimized). While I don't believe all high impact papers need to be 18+ % efficient, the mechanism by which the most efficient devices fail is important. Suggested improvements are stability tests that push the device to failure including light soaking under temperature and humidity followed by publication in a journal similar to E&ES.

Reply:

*As outlined above, we believe that the integration of the SnO_x layer inside the device is conceptually significantly different from earlier work. To be able to position a permeation barrier (SnO_x) inside the device stack, the SnO_x layer must be optically highly transparent **and** electrically conductive. As a part of the electron extraction layer, the SnO_x provides outstanding protection of the perovskite against the ingress of moisture and, even more importantly, at the same time it serves as powerful permeation barrier against the out-diffusion of decomposition products of the perovskite, and thereby, the overall decomposition of the perovskite is significantly suppressed.*

As the referee said, the efficiency of our cells is limited by the perovskite precursor used in this study. We agree with him that high impact papers do not necessarily need to report 18+ % efficient devices. The study of failure mechanisms of more efficient perovskite systems is certainly an important field, but it is beyond the scope of the present paper. Our paper focusses on a novel impermeable bi-layered electron extraction layer that serves as a diffusion barrier inside the perovskite solar cell. Thus, we decided to consider the most commonly used perovskite photo-active layer MAPbI₃. However, as stated in the manuscript, we believe that the concept of an impermeable electron extraction layer is generally applicable and therefore is expected to provide an avenue to achieve a substantially improved device lifetime of solar cells based on organo-metal halide perovskites even beyond MAPbI₃.

*As requested, we have updated the lifetime data for the cells aged at elevated temperature (60°C) which now extends to 1032 h (**new Figure 4**). We believe that these numbers now more impressively underline the stability enabled by our inverted device structure with impermeable AZO/SnO_x EELs.*

Figure. 4 Characteristics of inverted perovskite cells with varied cathode electron extraction assemblies (AZO/SnO_x/Ag, AZO/Ag) vs. time of storage at 60°C in nitrogen atmosphere. J_{sc} (a), V_{oc} (b), FF (c), and PCE (d). In this set of samples, the thickness of the ALD SnO_x layer was 20 nm and that of the Ag electrode was 100 nm. The characteristics have been determined from the J/V measurements in reverse direction (Figure S7).

We also agree with the referee that light soaking tests under concomitant heating are an important (and more demanding) stress condition. However, we want to note, that according to a very recent “Research Update” published by the Snaith group in *APL Materials* [*APL Mater.* 4, 091503 (2016)], the number of reports on the stability of perovskite solar cells in which the cells are actually stressed by light soaking **and** heat simultaneously, is very scarce. This may in part be due to the fact that upon “multi-stress” conditions the individual degradation mechanisms are difficult to be separated.

Nevertheless, as requested by the referees, we conducted a new set of long-term light soaking tests at 60°C. Briefly, the devices were illuminated with a white LED to achieve the same J_{sc} as upon AM1.5G solar irradiation. At the same time, the devices were kept in an

oven at 60°C under N₂ atmosphere (all experimental details are mentioned in the methods section). We have deliberately chosen the N₂ environment for the light-soaking experiment, as we had already clarified the influence of humidity in Figure 1, and we had clearly shown that the encapsulation due to the AZO/SnO_x layer is efficiently blocking the ingress of moisture.

The results of the new light-soaking/heating stress tests are shown in the new **Figure S8**.

In the manuscript we added the following discussion:

“Aside from so-called “mono-stress” conditions, like elevated temperature, multi-stressing can be considered. To this end, we conducted set of long-term light soaking tests at 60°C under N₂, in which the devices were simultaneously illuminated with a white LED to achieve the same J_{sc} as upon AM1.5G solar irradiation (for details see the experimental section). The results of this multi-stress experiment are shown in the supporting information (**Figure S8**). Briefly, again the AZO/SnO_x cells are substantially more stable than the AZO cells under concomitant heat and illumination. This difference is in part due to the suppressed decomposition of the perovskite due to heat in case of the AZO/SnO_x (in agreement with the discussion of **Figure 4**). However, there is a clear degradation even of the AZO/SnO_x samples where the PCE decays to roughly 60 % of its initial value after 300 hours. This is in contrast to the case where only thermal stress has been used. Earlier work has unraveled the photo-induced degradation of solar cells based on MAPbI₃. While Misra et al. identified a photo-activated decomposition mechanism², more recently Nie et al. reported that continuous illumination of MAPbI₃ caused the formation of trap states which spoiled the solar cells performance, especially J_{sc} .³ While our sealing approach based on the impermeable AZO/SnO_x electron extraction layer has been shown to efficiently suppress the decomposition of the perovskite, it cannot suppress the formation of trap states in the perovskite. Similar to the report of Nie et al., the cell characteristics recover after stressing, if the cells are kept in darkness (**Figure S8f**). This is why we conclude that photo-induced trap-state formation occurs in our cells. However, there have been recent reports confirming that photo-induced degradation is not a general problem of organo-lead halide perovskites, and very recently some optimized mixed cation / mixed halide perovskite active materials were shown to be far less prone to light soaking degradation.^{4,5} Heat was found to be a dominating source of degradation in these mixed cation perovskite cells.⁶ Note, our inverted device structure based on the impermeable AZO/SnO_x electron extraction layer is generally applicable and can also accommodate these next-generation perovskite photo-active systems with enhanced stability and efficiency.”

Figure S8 Characteristics of inverted perovskite cells with varied cathode electron extraction assemblies (AZO/SnO_x/Ag, AZO/Ag) vs. time of storage at 60°C in nitrogen atmosphere and simultaneous illumination with a white LED to achieve the same J_{sc} as under the solar simulator. J_{sc} (a), V_{oc} (b), FF (c), and PCE (d). $J_{sc,0}$, $V_{oc,0}$, FF_0 , and PCE_0 are the characteristics of the cells before stressing. In this set of samples, the thickness of the ALD SnO_x layer was 20 nm and that of the Ag electrode was 100 nm. The characteristics have been determined from the J/V

measurements in reverse direction. Spectrum of the white LED used for the light-soaking stress test under concomitant heating (e). J/V characteristics of an AZO/SnO_x/Ag cell, non-stressed (fresh), stressed for 300 hours (light/heat), and subsequently recovered in in darkness at room temperature for nine days (in the glove box).

Finally, we agree with the referee that Energy&Environmental Science (E&ES) would also be a potential outlet for our work. While we are convinced that both journals are highly respected, we decided to choose Nature Communications in this case.

Reviewer #2 (Remarks to the Author):

This manuscript described the use of a bi-layer oxide AZO/SnO_x as electron extraction layer to improve the stability of perovskite solar cell. Stability is a crucial issue for perovskite solar cell before it can be considered as a promising alternative for current existing technologies. The finding and phenomena reported in this communication could be interesting for the community. However, several issues need to be addressed before the acceptance by high-ranking journal.

1. The results of the most successful device on stability [Ref 17, Science 350, 944-948 (2015)] shall be briefly described and discussed in the introduction. The device architecture of this communication is conceptually similar to that Science paper, except the use of bilayer oxide. I think the advantages and novelty (vs that Science paper work) shall be clearly expressed in the introduction to show the importance of this work. Otherwise, changing materials can only be considered as a minor progression or follow-up.

Reply:

The Science paper mentioned by the referee used an inverted planar device structure. The authors employ NiMgLiO as hole extraction layer and Ti(Nb)O_x as electron extraction layer. Specifically, the Ti(Nb)O_x was prepared by a sol-gel process. It is known that sol-gel layers other than ALD layers do not form conformal pin-hole free coatings and thus they typically afford rather poor permeation barriers [see e.g. Hirvikorpi et al. Surface and Coatings Technology, 205, 5088 (2011)]. The SEM images in their supporting information also show some granularity of the layers. Grain-boundaries are pathways for gas diffusion. Opposed to that, our ALD grown SnO_x layers are amorphous and dense. In addition the authors had to limit the thickness of the sol-gel TiNbO_x to 10 nm due to its poor electrical conductivity of $10^{-5} \text{ S cm}^{-1}$. For thicker TiNbO_x layers some notable deterioration in the J/V characteristics is found. As already discussed in the original version of the paper, our ALD grown SnO_x layers show a more than two orders of magnitude higher electrical

conductivity of $5 \times 10^{-3} \text{ S/cm}$ which does not impose such a thickness limitation. Even though the authors did not provide any permeation data for their 10 nm thin Ti(Nb)O_x layers, for the reasons discussed above, it can be expected that the water permeation rate is substantially higher than that of ALD grown SnO_x layers with similar thickness. As result, the authors showed device stability only in a dry cabinet or with additional encapsulation of the devices. Opposed to that our devices proved stable in ambient air even without additional encapsulation. Likewise, even though the authors did not provide any experiments at elevated temperatures of 60°C , it must be expected that the 10 nm sol-gel TiNbO_x will not form a serious barrier against the egress of perovskite decomposition products. We have clearly shown that in our case the barrier functionality of the AZO/ SnO_x electron extraction layer not only provides outstanding protection of the perovskite against the ingress of moisture, but at the same time it serves as powerful permeation barrier against the out-diffusion of decomposition products of the perovskite, and thereby, the overall decomposition of the perovskite is significantly suppressed.

As suggested by the referee, we have added the following clear differentiation of our work compared to the report in Science 350, 944-948 (2015) in the introduction of our paper.

“Chen employed a sol-gel Ti(Nb)O_x electron extraction layer, which was aimed to protect the perovskite against moisture from ambient atmosphere.⁸ Due to its poor electrical conductivity of $10^{-5} \text{ S cm}^{-1}$, in devices the authors had to limit the thickness of the sol-gel Ti(Nb)O_x to 10 nm. Moreover, it is known that sol-gel derived layers state only relatively poor permeation barriers.⁹ As a result, the device stability tests which afforded a lifetime of 1000 h, were done with additional encapsulation.”

2. For the XRD results, the spectra seem to contain only the diffraction pattern of perovskite. As the author said in the manuscript the sample is a full device, I think the author shall see the diffraction of those constituent layers to prove that the sample is the full device and cover with EEL and metal contact. Please clarify this issue. It is quite unusual that under ambient condition, no PbI_2 signal were observed. In general the formation of PbI_2 prevails extremely fast in air. The explanation of "non-crystalline nature of the degradation product" is not very convincing as no amorphous feature in the XRD spectra.

Reply:

We fully understand the comment of the referee. To explain: The XRD spectra shown in Figure 3a were not measured with the uttermost sensitivity of our equipment, which means that (the weak) signals from Ag or ITO were not clearly visible. Even though, upon closer inspection of the spectrum in Figure 3a, a number of additional weak peaks are present. To clarify the question of the referee, we measured an identical sample again in XRD with a 10 times longer integration time (measurement time for one spectrum: 18 hours) to obtain a better signal to noise ratio and to render these weaker peaks better visible. Moreover we

chose a logarithmic representation and included this spectrum as new Figure S3 in the supporting information. Aside from the dominating MAPbI₃ reflections, we were able to clearly identify signals attributed to PbI₂, Ag and ITO. The AZO layer was not visible. AZO is a nano-crystalline material, however, the size of the AZO particles is only 12 nm, and therefore even in this more sensitive measurement we were not able to detect signs of the AZO layer. We have also prepared single AZO layers (80 nm thickness) on top of a glass substrate but we did also not see any XRD signal from these layers under similar XRD measurement conditions.

Figure S3: Logarithmic representation of the XRD spectrum of a fresh perovskite cell based on AZO as EEL with an assignment of the peaks to layers in the device. The peak positions of MAPbI₃,¹⁰ PbI₂,¹¹ ITO,¹² and Ag,¹³ were taken from the respective references.

We also updated Figure 3a and the inset in Figure 3a with the spectra measured with greater sensitivity. In the manuscript we now note the small PbI₂ peak that is present in the fresh samples, already. We also notice that after 7 days of storing the device in ambient air, the PbI₂ peak did not show a notable increase.

In the revised manuscript it now reads:

“As this XRD measurement has been made on a full device stack, further signals related to the Ag and ITO electrodes are visible. A detailed assignment of these peaks is shown in the supporting information (Figure S3). Only a very weak signal due to PbI₂ was found, which did not significantly increase even after seven days in air (Figure 3a). Thus, no severe

decomposition of the perovskite to PbI_2 upon storage in air can be inferred from the XRD data."

Moreover, we agree with the referee, that there is a general belief that $MAPbI_3$ exposed to air forms PbI_2 quite readily. However, a careful review of reports in the literature shows a more ambiguous picture. Especially, a more detailed specification is needed for "ambient conditions" or "exposure to air" (in terms of temperature and relative humidity). For example, Christians et al. have reported XRD studies of $MAPbI_3$ (fresh and after 28 days in air at $23\pm 1^\circ C$ and 50% relative humidity (similar to our lab)) and did not see any notable change in their XRD spectra [J. Am. Chem. Soc. 2015, 137, 1530–1538]. While at $23\pm 1^\circ C$ and 90% rH the same authors evidenced the formation of a hydrate phase clearly seen as additional peaks at 10.46° and 16.01°) after 7 days. In our case, at $23^\circ C$ and 50% relative humidity, we also did not see a notable growth of the PbI_2 peak even after 7 days (inset of Figure 3a). However, the detection of iodine at the surface (XPS data Figure 3b) and the severe degradation of the Ag electrode (Figure 3c,d) infer that iodine compounds leaked out of the perovskite. The more reasonable explanation is that the amount of PbI_2 that is formed upon storage in air is even smaller compared to the small amount of PbI_2 already present in the pristine perovskite layer. We note, however, that likely only very small amounts of halide containing decomposition products are required to degrade the interface of the Ag electrode to the device and lead to the deteriorated device characteristics

*We agree with the referee, that we cannot provide unambiguous evidence for **amorphous** PbI_2 , and thus we deleted this sentence.*

3. For the XPS results, the baseline of fresh devices made with SnO_x and AZO/ SnO_x shall be included. Since XPS is a surface technique that investigates on the first few 3~5 nm, I wonder if the perovskite can be detected if protected after 10 nm Ag and AZO. The authors shall provide the whole spectrum to show what they have detected on the surface. The signal of Eb close to 622 eV for the blue line on Fig 3 (b) shall be identified and discussed. Is it something other than AgI?

Reply:

As requested by the referee, we have added the baseline spectra of the fresh samples with SnO_x and AZO/ SnO_x in the revised Figure 3b.

We have also added the survey spectra for all samples in the supporting information (Figure S4). As expected, no signal due to the perovskite (e.g. $Pb4d$) can be detected, because the perovskite in these samples is covered by PCBM/AZO/Ag, or PCBM/ SnO_x /Ag, or PCBM/AZO/ SnO_x , respectively. Note, the position where the $Pb4f$ signals would be located is overlaid by $Zn3s$ and $Sn4s$ signals as indicated in the figure, thus we refer to the $Pb4d$ signal.

Figure S4: Survey photoemission spectra for cells based on AZO, SnO_x and bi-layered AZO/SnO_x EELs, respectively. Samples were fresh and aged for two days in air, respectively. The top Ag layer was only about 10 nm to see the metal-oxide underneath.

The peak around 622 eV in the XPS spectrum shown in Figure 3b, does not represent another bonding state of the iodine, it is rather a so called “shake-up” peak. The shake-up process is a two electron process in which a part of the kinetic energy of the photoelectron is used to lift-up an additional electron into an excited state, resulting in an excited ion. Due to the loss in kinetic energy, the apparent binding energy of the emitted and analyzed photoelectron is hereby increased by the energy necessary for the excitation (typically a few eV). Such shake-up peaks are commonly observed in any strong XPS signal. To show that the feature in Figure 3b is not due to an interaction of Iodine with e.g. silver, we analyzed the iodine XPS spectra of pure MAPbI₃, as well as of the precursors MAI and PbI₂ (see figure below). All of these spectra show this shake up feature at around 622 eV (red curve).

To clarify, we added in the revised version of the paper:

“The peak around 622 eV in the XPS spectrum of the aged AZO sample does not represent another bonding state of the iodine, it is rather a so called “shake-up” peak. Such shake-up peaks are commonly observed in strong XPS signals.”

4. Are the SEM images taken on top of Ag or on top of oxides? I think that matters for proposing the mechanism.

Reply:

The measurements were taken on top of Ag. It may have been overseen, but in the original version of the manuscript we had already specified this in the respective figure caption.

5. To demonstrate real stability for solar cells, light soaking is necessary. Especially for perovskite solar cell, light soaking results can really substantiate the claim for long term stability. To be published in such high impact journal like Nature Communications, I think the authors shall show stability under light soaking.

Reply:

*We also agree with the referee that light soaking tests (under concomitant heating) are an important (and more demanding) stress condition. However, we want to note, that according to a very recent “Research Update” published by the Snaith group in APL Materials [APL Mater. 4, 091503 (2016)], the number of reports on the stability of perovskite solar cells in which the cells are stressed by light soaking **and** heat simultaneously, is very scarce. This may in part be due to the fact that upon “multi-stress” conditions the individual degradation mechanisms are difficult to be separated.*

Nevertheless, as requested by the referees, we conducted a new set of long-term light soaking tests at 60°C. Briefly, the devices were illuminated with a white LED to achieve the same J_{sc} as upon AM1.5G solar irradiation. At the same time, the devices were kept in an oven at 60°C under N₂ atmosphere (all experimental details are mentioned in the methods section). We have deliberately chosen the N₂ environment for the light-soaking experiment, as we had already clarified the influence of humidity in Figure 1, and we had clearly shown that the encapsulation due to the AZO/SnO_x layer is efficiently blocking the ingress of moisture.

The results of the new light-soaking/heating stress tests are shown in the new Figure S8 (also shown above in this reply).

In the manuscript we added the following discussion:

“Aside from so-called “mono-stress” conditions, like elevated temperature, multi-stressing can be considered. To this end, we conducted set of long-term light soaking tests at 60°C under N₂, in which the devices were simultaneously illuminated with a white LED to achieve the same J_{sc} as upon AM1.5G solar irradiation (for details see the experimental section). The results of this multi-stress experiment are shown in the supporting information (Figure S8). Briefly, again the AZO/SnO_x cells are substantially more stable than the AZO cells under concomitant heat and illumination. This difference is in part due to the suppressed decomposition of the perovskite due to heat in case of the AZO/SnO_x (in agreement with the discussion of Figure 4). However, there is a clear degradation even of the AZO/SnO_x samples where the PCE decays to roughly 60 % of its initial value after 300 hours. This is in contrast to the case where only thermal stress has been used. Earlier work has unraveled the photo-induced degradation of solar cells based on MAPbI₃. While Misra et al. identified a photo-activated decomposition mechanism², more recently Nie et al. reported that continuous illumination of MAPbI₃ caused the formation of trap states which spoiled the solar cells performance, especially J_{sc} .³ While our sealing approach based on the impermeable AZO/SnO_x electron extraction layer has been shown to efficiently suppress the decomposition of the perovskite, it cannot suppress the formation of trap states in the perovskite. Similar to the report of Nie et al., the cell characteristics recover after stressing, if the cells are kept in darkness (Figure S8f). This is why we conclude that photo-induced trap-state formation occurs in our cells. However, there have been recent reports confirming that photo-induced degradation is not a general problem of organo-lead halide perovskites, and some optimized mixed cation / mixed halide perovskite active materials were shown to be far less prone to light soaking degradation.^{4,5} Heat was found to be a dominating source of degradation in these mixed cation perovskite cells.⁶ Note, our inverted device structure based on the impermeable AZO/SnO_x electron extraction layer is generally applicable and can also accommodate these next-generation perovskite photo-active systems with enhanced stability and efficiency.”

6. The unit of EQE on Figure 1b is wrong.

Reply:

Thank you for pointing us to this error. We have revised Figure 1 b accordingly.

7. The first inverted perovskite solar cell using PEDOT:PSS and PCBM shall be cited in the introduction. Adv. Mater. 2013, 25, 3727.

Reply:

As requested, we have added the reference to Adv. Mater. 2013, 25, 3727 in the introduction.

Overall, I suggest a minor revision for this manuscript.

Reply:

We deeply value your comments that helped us to improve our paper.

Reviewer #3 (Remarks to the Author):

The manuscript by Brinkmann et al address the use of AZO/SnO_x electron extractor layers in hybrid perovskite solar cells. More specifically, the paper reports improved device stability when such AZO/SnO_x layers are used in inverted type solar cells. The paper is interesting but in the current state the paper is not suitable for a journal like Nature Communications. There are a number of significant issues that must be addressed.

1 The more detailed mechanism of how the combined AZO/SnO_x layer interlayer improves stability needs to be established. This is not at all clear from the data presented in the paper.

2 Following from above, what is the mechanism by which the SnO_x layer improves stability? How does it block water getting in the device? Surely most of the diffusion occurs from the sides of the devices and through the silver? Why can't the metal electrode act as a barrier to water? It is not beyond the realms of possibility that approximately 100 nm thick (or several 10's nm thick) metal electrodes can function good physical blocking layers. Perhaps this should be investigated.

Reply:

Comments 1+2 point into the same direction, thus we are going to reply to them combined:

As detailed in the manuscript the root cause of the improved stability due to the addition of the SnO_x layer is in its functionality as a permeation barrier. The SnO_x provides outstanding protection of the perovskite against the ingress of moisture and, even more importantly, at the same time it serves as powerful permeation barrier against the out-diffusion of decomposition products of the perovskite, e.g. CH₃NH₃I, (CH₃)₃N, CH₃I, HI, etc. We showed that the sealing properties of the SnO_x layer contain the decomposition products inside the cell, and thereby, the overall decomposition of the perovskite is significantly suppressed. Water ingress via the sides (so-called: "side-leakage") is severely suppressed by thin-film barrier, as no glues are involved. Permeation via the Ag may occur but please note that the barrier is below the Ag (see Fig. 1a). The metal electrode can form some barrier against moisture. They are used for food packaging, e.g. for potato-chips bags. But as is known, metal films do not form the kind of barriers that are required to protect sensitive thin-film devices such as OLEDs or, here, perovskite solar cells. To give some numbers: The water vapor transmission rate (WVTR) through 300 nm thick metal layers has been determined, see e.g. Appl. Phys. Lett. 93, 133307 (2008), to be on the order of 0.2 g/(m² day), while the ALD grown SnO_x layers provide WVTRs on the order of 7x10⁻⁵ g/(m² day). Thus, just like in the case of OLEDs and organic solar cells, the top metal electrode will not provide the level of protection required. Special to perovskites: There is the intrinsic decomposition of CH₃NH₃PbI₃ to CH₃NH₃I and PbI₂, which is thermally activated and which occurs even under inert conditions. The corrosion of metal electrodes due to these decomposition products has been identified to be a critical issue. Thus a permeation barrier against the out-diffusion of decomposition products of the perovskite is required. As we show, the sealing properties of the SnO_x layer contain the decomposition products inside the cell, and thereby, the overall decomposition of the perovskite is significantly suppressed and the deterioration of the electrode is prevented.

3 The authors suggest that the silver electrode may be involved in the degradation of the device in presence of water. Have the authors tried to replace the silver electrode with lets say gold (which is known to be more stable than silver) in control devices that do now contain the SnO_x layer: ITO/PEDOT/active layer/gold.

Reply:

We are aware of some reports claiming improved stability of perovskite solar cells due to the use of Au. On the other hand, this claim is not free of ambiguity, as there are also reports that found long-term stability issues related to the migration of Au.⁶ While we assume that our impermeable AZO/SnO_x electron extraction layer could be well suited to also prevent the migration of electrode metal atoms into the active layer, we feel that this is beyond the scope of the present communication. A further reason why we did not consider Au is its high cost, which might render Au electrodes less attractive for future commercialization.

Note: Point 4 was missing in the original referee report!

5 The authors should provide more information about how the device stability experiments were performed. Was this done with maximum power point tracking? Were the experiments performed whilst continuously illuminating the solar cells under load with the solar cells put in ambient environment? Or, are the authors just referring "shelf life" stability. To get a true indication of stability it is important that the authors consider the former (e.g.) measurements under continuous illumination under load with MPP tracking.

Reply:

We applied so-called "mono-stress" conditions, i.e. either in air or under N₂ at 60°C without light soaking.

*Nevertheless, as requested by the referees, we conducted a new set of long-term light soaking tests at 60°C. However, we want to note, that according to a very recent "Research Update" published by the Snaith group in APL Materials [APL Mater. 4, 091503 (2016)], the number of reports on the stability of perovskite solar cells in which the cells are stressed by light soaking **and** heat simultaneously, is very scarce. This may in part be due to the fact that upon "multi-stress" conditions the individual degradation mechanisms are difficult to be separated.*

Briefly, our devices were illuminated with a white LED to achieve the same J_{sc} as upon AM1.5G solar irradiation. At the same time, the devices were kept in an oven at 60°C under N₂ atmosphere (all experimental details are mentioned in the methods section). We have deliberately chosen the N₂ environment for the light/heat stress experiment, as we had already clarified the influence of humidity in Figure 1, and we had clearly shown that the encapsulation due to the AZO/SnO_x layer is efficiently blocking the ingress of moisture.

The results of the new light-soaking/heating stress tests are shown in the new Figure S8 (also shown above in this reply).

In the manuscript we added the following discussion:

"Aside from so-called "mono-stress" conditions, like elevated temperature, multi-stressing can be considered. To this end, we conducted set of long-term light soaking tests at 60°C under N₂, in which the devices were simultaneously illuminated with a white LED to achieve the same J_{sc} as upon AM1.5G solar irradiation (for details see the experimental section). The results of this multi-stress experiment are shown in the supporting information (Figure S8). Briefly, again the AZO/SnO_x cells are substantially more stable than the AZO cells under concomitant heat and illumination. This difference is in part due to the suppressed decomposition of the perovskite due to heat in case of the AZO/SnO_x (in agreement with the discussion of Figure 4). However, there is a clear degradation even of the AZO/SnO_x samples where the PCE decays to roughly 60% of its initial value after 300 hours. This is in contrast to the case where only thermal stress has been used. Earlier work has unraveled the photo-induced degradation of solar cells based on MAPbI₃. While Misra et al. identified a photo-activated decomposition mechanism², more recently Nie et al.

reported that continuous illumination of MAPbI₃ caused the formation of trap states which spoiled the solar cells performance, especially J_{sc}.³ While our sealing approach based on the impermeable AZO/SnO_x electron extraction layer has been shown to efficiently suppress the decomposition of the perovskite, it cannot suppress the formation of trap states in the perovskite. Similar to the report of Nie et al., the cell characteristics recover after stressing, if the cells are kept in darkness (**Figure S8f**). This is why we conclude that photo-induced trap-state formation occurs in our cells. However, there have been recent reports confirming that photo-induced degradation is not a general problem of organo-lead halide perovskites, and some optimized mixed cation / mixed halide perovskite active materials were shown to be far less prone to light soaking degradation.^{4,5} Heat was found to be a dominating source of degradation in these mixed cation perovskite cells.⁶ Note, our inverted device structure based on the impermeable AZO/SnO_x electron extraction layer is generally applicable and can also accommodate these next-generation perovskite photo-active systems with enhanced stability and efficiency.⁷

6 In the temperature stability measurements shown in figure 4. It is shown the drop in performance is mainly due to a drop in the fill factor. The authors should discuss the origin of this. The authors should also perhaps repeat temperature stability experiments using a thicker electrode layer - certainly thicker than 10nm !! It is not surprising that with such thin layers of silver (e.g. 10nm) increase the chance that the degradation products escape.

Reply:

Obviously, there is a misunderstanding. The devices presented in Figure 4 were based on 100 nm thick silver electrodes. The 10 nm Ag layers were only used in the XPS experiments to provide more sensitivity in the degradation effect.

To avoid further misconceptions we have specified the electrode thickness in the caption of Figure 4:

"In this set of samples, the thickness of the ALD SnO_x layer was 20 nm and that of the Ag electrode was 100 nm."

Regarding the origin of the loss in FF in Figure 4:

We discussed in the manuscript:

*"Under inert atmosphere, the thermally activated degradation of the perovskite leads to decomposition products that can easily diffuse through the PCBM/AZO/Ag layers on top of the perovskite. The less volatile PbI₂ phase remains in the active layer leading to a substantial decay of the FF of the corresponding devices (**Figure 4c**). Moreover, it has been shown earlier that a change in film stoichiometry (Pb-poor to Pb-rich) may substantially alter the electronic properties of the perovskite and especially the position of the conduction and valence band with respect to the vacuum level.¹⁴ Towards increasingly Pb-rich conditions, the ionization energy of the perovskite has been shown to increase*

significantly to values in the order of $> 6\text{eV}$, which would render the extraction of holes more and more challenging."

Again, we would like to thank the referees for their very helpful comments. We hope that in the revised version, our paper has now become acceptable for *Nature Communications*.

Thank you for your time and efforts!

Sincerely,

Thomas Riedl

1. Kim IS, Martinson ABF. Stabilizing hybrid perovskites against moisture and temperature via non-hydrolytic atomic layer deposited overlayers. *Journal of Materials Chemistry A* **3**, 20092-20096 (2015).
2. Misra RK, *et al.* Temperature- and Component-Dependent Degradation of Perovskite Photovoltaic Materials under Concentrated Sunlight. *The Journal of Physical Chemistry Letters* **6**, 326-330 (2015).
3. Nie W, *et al.* Light-activated photocurrent degradation and self-healing in perovskite solar cells. *Nat Commun* **7**, 11574 (2016).
4. Saliba M, *et al.* Incorporation of rubidium cations into perovskite solar cells improves photovoltaic performance. *Science*, (2016).
5. McMeekin DP, *et al.* A mixed-cation lead mixed-halide perovskite absorber for tandem solar cells. *Science* **351**, 151-155 (2016).
6. Domanski K, *et al.* Not All That Glitters Is Gold: Metal-Migration-Induced Degradation in Perovskite Solar Cells. *Acs Nano* **10**, 6306-6314 (2016).
7. Liu J, *et al.* High-Quality Mixed-Organic-Cation Perovskites from a Phase-Pure Non-stoichiometric Intermediate (FAI) $_{1-x}$ PbI $_2$ for Solar Cells. *Adv Mater* **27**, 4918-4923 (2015).
8. Chen W, *et al.* Efficient and stable large-area perovskite solar cells with inorganic charge extraction layers. *Science* **350**, 944-948 (2015).
9. Hirvikorpi T, Vähä-Nissi M, Nikkola J, Harlin A, Karppinen M. Thin Al $_2$ O $_3$ barrier coatings onto temperature-sensitive packaging materials by atomic layer deposition. *Surface and Coatings Technology* **205**, 5088-5092 (2011).

10. Safdari M, Fischer A, Xu B, Kloo L, Gardner JM. Structure and function relationships in alkylammonium lead(ii) iodide solar cells. *Journal of Materials Chemistry A* **3**, 9201-9207 (2015).
11. Zhou Y, *et al.* Growth control of compact CH₃NH₃PbI₃ thin films via enhanced solid-state precursor reaction for efficient planar perovskite solar cells. *Journal of Materials Chemistry A* **3**, 9249-9256 (2015).
12. Sunde TOL, *et al.* Transparent and conducting ITO thin films by spin coating of an aqueous precursor solution. *J Mater Chem* **22**, 15740-15749 (2012).
13. Kato Y, Ono LK, Lee MV, Wang S, Raga SR, Qi Y. Silver Iodide Formation in Methyl Ammonium Lead Iodide Perovskite Solar Cells with Silver Top Electrodes. *Advanced Materials Interfaces* **2**, 1500195 (2015).
14. Emara J, Schnier T, Pourdavoud N, Riedl T, Meerholz K, Olthof S. Impact of Film Stoichiometry on the Ionization Energy and Electronic Structure of CH₃NH₃PbI₃ Perovskites. *Adv Mater* **28**, 553-559 (2016).

Dear Editor,

Thank you for your notification regarding our revised submission NCOMMS-16-14487A. Here is our point-by-point reply to the referees (please note our answers are in *italic*):

Reviewer #1 (Remarks to the Author):

I am satisfied with the revisions to the manuscript. Although a simultaneous temperature and moisture (damp heat) test, which all commercial PV must pass, would be most interesting there is sufficient data included here for an initial report. The changes raise the quality of this revised work to those required by Nature Comm. I support prompt publication.

We value the efforts of the referee and appreciate his supportive recommendation

Reviewer #2 (Remarks to the Author):

The revised version has addressed most of the comments and questions raised by the reviewer properly. I would recommend its publication.

We value the efforts of the referee and appreciate his supportive recommendation

Reviewer #3 (Remarks to the Author):

Based on the responses and reading the revised paper the referee believes that the paper may now be suitable for publication in Nature communications.

Point 1 and 2: The role of metal contact versus SnOx layer in protecting against water vapour ingress has been clarified. The comparison between vapour transmission rates for metal versus ALD grown SnOx is welcome.

Point 3: Though a comparison of gold and silver as an electrode and the investigation of the use of SnOx/AZO as a blocking layer (ie reducing migration of metal atoms from electrode) are important issues the referee accepts that this is potentially beyond the scope of the current work.

Point 5: The additional solar cell stability experiments (Figure S8) are welcome and serve to clarify the concerns of referee 3.

Point 6: comments regarding thickness of Ag layer clarified. Concerning the origin of the drop in the fill factor (FF): the discussion is welcome. However, the referee has an additional question here: Did the authors observe a change in hysteresis in fresh versus aged samples. (referring to a device that is treated in the same way as in Figure 5; main manuscript). For example, a device that is using an AZO/SnOx extraction layer (with and without heating at 60C)?

Reply:

We value the efforts of the referee and appreciate his supportive recommendation

We added in the manuscript: "Note, the hysteresis did not increase in the course of aging."

Again, we would like to thank everyone involved in handling and reviewing our paper. We are looking forward to seeing our paper published in *Nature Communications*.

Sincerely,

Thomas Riedl